# Association of Smoking and Respiratory Disease History with Pancreatic Pathologies Requiring Surgical Resection

**DOI:** 10.3390/cancers15112935

**Published:** 2023-05-26

**Authors:** Carolyn Ream, Matthew Sabitsky, Rachel Huang, Emma Hammelef, Theresa P. Yeo, Harish Lavu, Charles J. Yeo, Wilbur Bowne, Avinoam Nevler

**Affiliations:** 1Sidney Kimmel Medical College, Thomas Jefferson University, Philadelphia, PA 19107, USA; carolyn.ream@students.jefferson.edu (C.R.); matthew.sabitsky@students.jefferson.edu (M.S.); emma.hammelef@students.jefferson.edu (E.H.); 2Jefferson Pancreas, Biliary and Related Cancer Center, Department of Surgery, Philadelphia, PA 19107, USA

**Keywords:** pancreatic tumor, pancreatic cancer, pancreatic ductal adenocarcinoma (PDAC), intraductal papillary mucinous neoplasm (IPMN), chronic obstructive pulmonary disease (COPD), obstructive sleep apnea (OSA), asthma, smoking, respiratory diseases, hypercapnia, hypoxia

## Abstract

**Simple Summary:**

Smoking, chronic obstructive pulmonary disease and obstructive sleep apnea have been recognized as risk factors for the development of cancer. We aimed to analyze the association between patients’ respiratory background and various pancreatic pathologies in patients that underwent a curative-intent resection of the pancreatic head. Using retrospective data from a prospectively maintained database sourced from a high-volume single center, smoking was identified to be strongly associated with pancreatic cancer and inflammatory pancreatic masses. A gender-specific association was found in males between chronic obstructive pulmonary disease and intraductal papillary mucinous neoplasms (IPMNs). In females, an association was found between obstructive sleep apnea and IPMNs. Given these multiple associations between hypercapnic and hypoxic respiratory diseases and premalignant and malignant pancreatic lesions, we propose that the pancreatic respiratory microenvironment may play a role in the incidence of certain pancreatic lesions.

**Abstract:**

Background: The purpose of this study was to examine the relationship between various respiratory conditions, including hypercapnic respiratory disease, and a multitude of resected pancreatic lesions. Methods: This retrospective case-control study queried a prospectively maintained database of patients who underwent pancreaticoduodenectomy between January 2015 and October 2021. Patient data, including smoking history, medical history, and pathology reports, were recorded. Patients with no smoking history and no concomitant respiratory conditions were designated as the control group. Results: A total of 723 patients with complete clinical and pathological data were identified. Male current smokers showed increased rates of PDAC (OR 2.33, 95% CI 1.07–5.08, *p* = 0.039). Male patients with COPD had a markedly increased association with IPMN (OR 3.02, CI 1.08–8.41, *p* = 0.039), while females with obstructive sleep apnea had a four-fold increase in risk of IPMN compared to women in the control group (OR 3.89, CI 1.46–10.37, *p* = 0.009). Surprisingly, female patients with asthma had a decreased incidence of pancreatic and periampullary adenocarcinoma (OR 0.36, 95% CI 0.18–0.71. *p* < 0.01). Conclusion: This large cohort study reveals possible links between respiratory pathologies and various pancreatic mass-forming lesions.

## 1. Introduction

Primary pancreatic lesions can be broadly classified into exocrine, endocrine, mixed and other. These include pancreatic ductal adenocarcinoma (PDAC), periampullary carcinoma, cholangiocarcinoma, mucinous and serous cysts, intraductal papillary mucinous neoplasm (IPMN), pancreatic neuroendocrine tumors (PNET), and others. Recent studies have demonstrated the impact of hypoxia and the hypoxia-signaling pathway in tumor formation and progression [1,2,3]. Even in the pancreas specifically, increased hypoxia signaling has been noted across multiple pancreatic neoplasm types [4].

The relationship between pancreatic cancer and smoking is well established, as meta-analyses have indicated a 1.7-fold increase in relative risk of pancreatic cancer in patients with a positive smoking history [5,6,7]. However, beyond smoking, chronic obstructive pulmonary disease is also known to be a risk factor for extrapulmonary cancers. Multiple studies have found chronic obstructive pulmonary disease (COPD) to be an independent risk factor for the development of cancer overall and pancreatic cancer specifically [8,9,10]. The respiratory microenvironment and carcinogenic impact of tobacco smoke have also been known to impact the progression of non-malignant pancreatic lesions to malignant pancreatic cancer. Intraductal mucinous papillary neoplasms, which are precursor lesions with a high malignant potential, were noted to be more prone to progress to malignancy in patients with a positive smoking history [10,11,12].

COPD is not the only respiratory disease that seems to be associated with increased cancer risk; other hypoxic-hypercapnic respiratory disorders, such as obstructive sleep apnea (OSA), have also been linked to an increased risk of cancer [13,14,15]. In pancreatic cells specifically, animal models have elucidated possible mechanisms for pancreatic cellular damage in an OSA-like respiratory microenvironment. Animal models of OSA have been shown to result in a reduced insulin:proinsulin ratio in pancreatic tissue, formation of pancreatic lesions, and pancreatic cellular apoptosis in a hypoxia dose-dependent manner [1]. Additionally, hallmarks of respiratory disease have been found to be associated with chronic pancreatitis, a known risk factor for pancreatic cancer and a significant pathology in and of itself. Decreased pulmonary diffusion capacity (DLco and DL/VA), which is frequently associated with emphysema and interstitial lung disease, was found by Masoero et al. to be highly associated with chronic calcific pancreatitis [16].

Given the existing data suggesting a complex relationship between the respiratory microenvironment and multiple pancreatic pathologies treatable with surgical resection, this study was initiated to further characterize possible associations and hypothesize mechanisms. This study aimed to assess the respiratory background in a large single-institutional cohort of patients undergoing pancreatic resection and to determine potential associations between pulmonary conditions and specific pancreatic pathologies.

## 2. Materials and Methods

### 2.1. Data Collection

This was a retrospective case-control study. The data were retrieved from a prospectively maintained database of patients who underwent pancreaticoduodenectomy at the Jefferson Pancreas, Biliary, and Related Cancer Center between 2013 and 2021. Data were collected on patients’ smoking history, past medical history, pathology, and clinical outcomes, among other information. Electronic medical records were reviewed to determine smoking history and identify patients that had a documented diagnosis of COPD, asthma, or OSA.

### 2.2. Cohort Selection

The original cohort was composed of 1093 patients. Due to insufficient perioperative data, 266 patients from the years 2013 and 2014 were excluded. Seventy-four patients were then excluded from this cohort of 827 based on incomplete or inconclusive pathological data. Of these 753 patients, 30 patients were excluded based on incomplete demographic information, yielding a final cohort of 723 patients (Figure 1).

### 2.3. Statistical Analysis

A control group was formed comprising patients that had neither a documented history of smoking nor a diagnosed respiratory disease (NRD). Statistical analysis included chi-square analyses of categorical data and Spearmen’s correlations of continuous data. SPSS (Version 28.0.1) was used to compare each respiratory condition to this control group. The *p*-values less than 0.05 were considered to be significant.

### 2.4. Ethical Approval

This study was performed at Thomas Jefferson University Hospital (TJUH) and approved by the local institutional review board. The data supporting the findings of this study are available from the corresponding author upon request.

## 3. Results

### 3.1. Cohort Description

The final cohort included 723 patients (see Table 1). The mean age was 66.8 years (±11.25), and the mean BMI was 27 (±5.6). The cohort was nearly equally split between males and females. Of our cohort, 83% were White/Caucasian, and 9% were Black/African American. A total of 382 (52.8%) patients had a positive smoking history, and 25.2% had a reported chronic respiratory condition, with 8.2% having COPD, 10.8% having asthma, 11.3% having obstructive sleep apnea, and 4.7% having more than one respiratory condition. The main pathologies encountered in this cohort of head-of-pancreas resections included 374 (52%) patients with PDAC, 79 (11%) patients with IPMN, 53 (7%) patients with periampullary adenocarcinoma, 50 (7%) patients with pancreatic neuroendocrine neoplasms, and 21 (3%) patients with pathologically documented chronic pancreatitis (see Table 2).

### 3.2. Distribution of Respiratory Diseases

Of the entire cohort, 267 (37%) patients had no respiratory disease or smoking history (NRD), and this group served as the control group for all statistical analyses performed. Thirty-eight percent of the cohort had a history of past or current smoking with no obstructive lung disease, and 15% of the cohort had both a history of smoking and a diagnosis of at least one chronic lung disease. Ten percent of patients had a chronic lung disease with no documented history of smoking.

### 3.3. Smoking Is Linked with Pancreatic Pathologies

Smoking was found to be closely associated with the male sex, with male patients being more likely to be a current or former smoker (OR 1.6, 95% CI 1.2–2.2, *p* < 0.01). Current smokers were strongly associated with histologically proven chronic pancreatitis (OR 4.46, CI 1.08–18.34, *p* = 0.046). Additionally, heavy smokers (current and former smokers with more than 40 pack years) showed an even greater incidence of chronic pancreatitis (OR 5.48, 95% CI 1.43–21.02. *p* = 0.016) (Figure 2). Male current smokers also showed significantly increased rates of PDAC (OR 2.33, 95% CI 1.07–5.08, *p* = 0.039).

### 3.4. Respiratory Diseases and Linkage to Pancreatic Lesions

The analysis of the COPD patients compared to the NRD group showed COPD to be closely associated with older age (69.9 vs. 66.3 years, *p* < 0.01) and the male sex (OR 2.0, 95% CI 1.13–3.55, *p* = 0.02). Overall (including both male and female patients), there were no statistical correlations between COPD and pancreatic pathologies. However, in a sex-based subgroup analysis, male patients with COPD had a markedly increased association with IPMN (OR 3.02, CI 1.08–8.41, *p* = 0.039) as compared to patients from the NRD group. Additionally, though borderline significant, in patients that were diagnosed with pancreatic pathology at the age of 65 years or younger, COPD also correlated with IPMN (OR 4.4, 95% CI 0.98–19.79, *p* = 0.07). Further statistical analyses comparing the risk of IPMN between severe COPD (steroid-dependent) and mild COPD (non-steroid-dependent) did not reveal any significant impact of disease severity on IPMN risk.

Compared to patients with no respiratory disease, obstructive sleep apnea was significantly correlated with higher BMI (26.25 ± 5.26 vs. 29.93 ± 5.95, *p* < 0.01) and the male sex (OR 3.57, 95%CI 2.09–6.10, *p* < 0.01). BMI itself was not found to be associated with IPMN (P = NS). Patients with obstructive sleep apnea had a decreased incidence of periampullary adenocarcinoma (OR 0.28, 95%CI 0.08–0.93, *p* = 0.03) as compared with the NRD group). Females with obstructive sleep apnea had a four-fold increase in risk of IPMN as compared to women in the NRD group (OR 3.89, CI 1.46–10.37, *p* = 0.009) (Figure 3a). Male patients with OSA did not have a difference in risk of IPMN (OR 0.79, CI 0.25–2.40, *p* = 0.772) (Figure 3b).

Though borderline in terms of statistical significance, asthma patients in the cohort tended to be younger than patients without asthma (64.6 ± 11.9 vs. 67.0 ± 11.2 years, *p* = 0.066), and female (OR 0.6, 95% CI 0.4–1.0, *p* = 0.055). Female patients with asthma were more likely to have pancreatic neuroendocrine tumors (OR 3.2, 95% CI 1.2–8.2, *p* = 0.02) and a decreased incidence of pancreatic and periampullary adenocarcinoma (OR 0.36, 95% CI 0.18–0.71. *p* < 0.01) (Figure 4a). These findings were not identified in males in our cohort, as male patients with asthma did not have a difference in risk of pancreatic and periampullary adenocarcinoma (OR 1.62, CI 0.96–3.76, *p* < 0.31) (Figure 4b). Analyses comparing rates of pancreatic and periampullary adenocarcinoma among patients with varying severity levels of asthma were inconclusive.

## 4. Discussion

The pancreatic respiratory microenvironment and the hypoxia signaling system plays an intriguing role in the initiation and progression of pancreatic neoplasms. Key pathway proteins such as Hypoxia-inducible factor-1 and Hypoxia-inducible factor-2 have been associated with carcinogenesis and solid tumor progression. A recent histologic assessment of pancreatic lesions has also shown significantly increased levels of these target proteins in IPMNs, neuroendocrine lesions and in PDAC [17]. In light of the importance of the tissue-specific respiratory microenvironment, we aimed to assess the impact of systemic respiratory conditions on the incidence of pancreatic pathologies.

Similar to studies by Heinen et al. [6] and Molina-Montes et al. [7], our study found an association between a history of smoking and the development of pancreatic and periampullary adenocarcinoma, though this finding was only significant in male patients. This finding could be explained by the association between smoking with the male sex in our cohort, yielding a cohort with an enriched number of male smokers as compared to female smokers. Pancreatic cancer appears to have a dose-dependent correlation with smoking [7]. This mechanism is further complicated by genetic variants in genes involved in the cellular glutathione redox system such as GSTM1, which have been associated with an increased risk of pancreatic cancer in those individuals that smoke [18].

Our current study also observed a relationship in which current smokers had an increased incidence of histologically proven chronic pancreatitis. Similarly, heavy smokers had an even greater rate of pancreatitis (Figure 2), suggesting that there is not only a potential relationship between smoking and chronic pancreatitis, but that this relationship might also show dose-dependency. Possible mechanisms previously described for the relationship between smoking and pancreatitis include nicotine-induced damage to acinar cells of the pancreas via signal transduction pathways, altered gene expression of the proteolytic enzyme trypsinogen and its inhibitor, or hypoxia-mediated mechanisms that promote pancreatic damage [1,19,20].

There may be some confounding effects in our analysis from alcohol consumption in patients with chronic pancreatitis. Alcohol is a well-known risk factor for chronic pancreatitis [21], and there is a known correlation between smoking and increased alcohol consumption [22,23]. Additionally, smoking has been shown to have a synergistic damaging effect on the pancreas when combined with alcohol in patients with non-gallstone pancreatitis, resulting in more severe and frequent pancreatitis [21,24,25].

However, evidence from the past decades have shown that smoking itself is also an independent risk factor for chronic pancreatitis and idiopathic pancreatitis [26,27,28,29,30]. In the context of our specific study cohort, as the pancreatitis patients underwent their resection due to a mass-forming pancreatic lesion, the correlation between smoking and idiopathic pancreatitis seems especially fitting.

Similar to the smoking-related increased risk of cancer, COPD has been identified as an independent risk factor for the development of cancers, even when controlling for smoking history [8,9]. One of our goals in this study was to determine if the associations between hypoxic/hypercapnic respiratory pathologies and increased cancer incidence holds true for pancreatic cancer. We have found a correlation between COPD and the co-occurrence of intraductal papillary mucinous neoplasms in males. IPMNs are common precursor lesions to pancreatic cancer that, if left untreated, can progress to invasive PDAC. Similar to smoking, COPD was associated with the male sex, and this is congruent with existing studies pointing to the male sex and smoking history as risk factors for the early detection of IPMN [11,31], especially in the context of our cohort of resected patients—which has the inherent selection bias of patients’ medical fitness for surgery. Several possible mechanisms may be involved in this correlation. First, we should recognize the known inflammatory and carcinogenic effects of smoking on human physiology and cellular biology. These effects are also closely linked to the development of COPD. However, we should also take into consideration that 12–35% of COPD patients have no previous history of smoking, which may suggest that some other mechanisms are at play [32,33,34,35,36]. Overall, our findings are, in part, compatible with large-scale epidemiologic data from Kornum et al. and Chiang et al. that suggest increased pancreatic cancer rates in COPD patients [9,10].

Obstructive sleep apnea has also been previously implicated as a cancer risk factor. Both Brenner et al. [15] and Palamaner et al. [37] measured all-cancer incidence and found an increased all-cancer incidence in patients with obstructive sleep apnea. Chang, et al. have prospectively followed a cohort of 846 women with sleep apnea (with a 1:5 age-matched controls) and reported a twofold increased risk for breast cancer in the OSA patients [38]. Another large study based on Taiwan’s National Health Institute (NHI) data found that sleep apnea is associated with a 1.54-fold increased risk in CNS cancers [39]. However, a specific relationship between OSA and pancreatic cancer incidence specifically has yet to be documented. Our study revealed that females with obstructive sleep apnea had a four-fold increase in risk of IPMN compared to women in the NRD group, a finding consistent with previous studies on OSA and increased cancer incidence. Interestingly, a previous study on resected pancreatic ductal adenocarcinomas showed that OSA diagnosis prior to surgery correlated with lower TNM staging and increased likelihood of having negative lymph node metastasis; however, clinical management and survival rates were no different between patients with and without OSA [40]. Even when controlling for node status, OSA patients have similar survival rates compared to patients without OSA, suggesting that other factors such as margin status and chemotherapy response may be more important prognostic indicators [40]. While the hypoxic tumor microenvironment has been linked to immunosuppression and oncogenesis, it is unclear why patients with OSA are less likely to have nodal involvement, although this may not be important given the similar clinical outcomes compared to non-OSA patients.

COPD has been shown to promote immune dysregulation in part through dysfunction via increased activity of regulatory T-cells and myeloid-derived suppressor cells [41,42,43]. Similarly, obstructive sleep apnea has been shown to induce a systemic pro-inflammatory state and has been linked with endothelial injury and atherosclerosis [44]. These effects have been identified to strongly stem from activation of the hypoxia signaling system [45]. Hypoxia inducible factor 1α (HIF-1α) plays a critical role in angiogenesis, metabolic reprogramming, cell replication, metastasis and cancer stem cell renewal in response to hypoxia [46,47,48,49,50,51,52]. Fujino et al. [45] showed that the expression of HIF-1α correlated with poor disease-free survival and overall survival in pancreatic neuroendocrine tumors. Animal models of OSA that mimicked intermittent hypoxia have been shown to result in pancreatic endocrine dysfunction, formation of pancreatic lesions, and pancreatic cellular apoptosis [1]. The hypoxia signaling pathway offers therapeutic targets as well, and the HIF inhibitor belzutifan has recently been approved by the FDA for patients with upstream VHL (von Hippo-Lindau) mutations who have pancreatic neuroendocrine tumors [53].

Similar to hypoxia, hypercapnia has also been implicated in cancer growth and cancer cell aggressiveness. Obata et al. [54] found that exposure of colon cancer cells to hypercapnic atmospheres promoted cancer invasiveness and the expression of matrix metalloproteinases. Other studies have found that hypercapnia induces chemotherapy resistance in lung cancer cells [55]. We have previously found that PDAC cells exposed to hypercapnic environments replicate more rapidly and become more resistant to radiotherapy and oxaliplatin therapy [2,56]. Overall, these studies suggest an important role for the respiratory microenvironment, which may explain the observed associations we have found in our study.

Despite previous studies suggesting increased cancer risk in patients with respiratory diseases such as COPD and emphysema, asthma may be an *exception* to this observation. The results from our study suggest a *negative* association between asthma and pancreatic cancer in female patients, with a more than *twofold decrease* in PDAC rates in female patients with a history of asthma. The limitation of this result to female patients in our cohort may be due to the borderline association of asthma with the female sex in our cohort. This finding of reduced PDAC risk in asthma patients is surprising in light of the previously published associations between asthma and increased cancer risk of the lung, stomach, liver, prostate, hematologic, colorectal, endometrial, ovarian, and other cancers [57,58]. However, pancreatic cancer may indeed be an exception to this trend, as several previous reports have noted that asthma and other atopic disorders have a protective association with PDAC [59]. Our findings are therefore congruent with the findings reported by Gomez-Rubio et al. [60,61] showing an inverse relationship between asthma diagnosis and the incidence of pancreatic ductal adenocarcinoma. The mechanism for this observed relationship is unclear, and further studies are required to identify the exact mechanism of this phenomenon. However, several possible mechanisms have been proposed. The first is based on the observation that patients with asthma that are actively taking medications for their condition showed significant PDAC risk reduction, which suggests that one or more asthma medications (such as cromoglicic acid used to stabilize mast cells) have some protective effect against PDAC [60,62]. A second proposed mechanism is the presence of a hyperactive immune state in patients with asthma, which results in sustained elevated IgE levels. This hyperactive immune state, while resulting in pathologic respiratory disease, may protect the patient from neoplastic processes that result in pancreatic cancer. Thirdly, a study by Cotterchio et al. investigating atopy-associated single nucleotide polymorphisms (SNPs) found that eighteen SNPs located in fourteen genes were associated with pancreatic cancer risk, with two SNPs in the LRP1B gene found to be associated with a decreased risk of pancreatic cancer after adjusting for multiple comparisons [63]. This gene has been implicated in antigen presentation, inflammation, and cancer progression, providing a potential basis for its association with both asthma and cancer risk.

## 5. Limitations

This study has several limitations that merit discussion. First, this single-institution cohort only included resectable pancreatic lesions of the pancreatic head, and therefore, associations could be missed between respiratory diseases and advanced cases of pancreatic cancer or left-sided pancreatic lesions. This selection bias needs to be considered when trying to generalize the significance of our findings. This study also had low rates of rarer pathologies such as solid pseudopapillary neoplasms and acinar cell carcinomas, and therefore, correlations between these pancreatic lesions and respiratory pathologies could not be analyzed, despite our relatively large cohort. Our assessment of the association between smoking and pancreatitis was unable to take into account the effects of alcohol intake as a cause for chronic pancreatitis resulting in pancreatic resection; finally, the retrospective nature of this study lends itself to possible bias and limits the determination of causality versus mere association. Nonetheless, as our cohort was based on a prospectively maintained database from a large urban hospital that serves a diverse patient population, we believe it offers meaningful reliability and represents our resected patient population.

## 6. Conclusions

The tumor microenvironment has been shown to play an important role in pancreatic neoplasia. In this study we show important links between respiratory disorders and the development of pancreatic pathologies requiring surgical resection. These results also support the findings of recent studies that demonstrate an unexplained association of asthma with decreased rates of pancreatic cancer, which will require further investigations to more fully characterize.

## Figures and Tables

**Figure 1 cancers-15-02935-f001:**
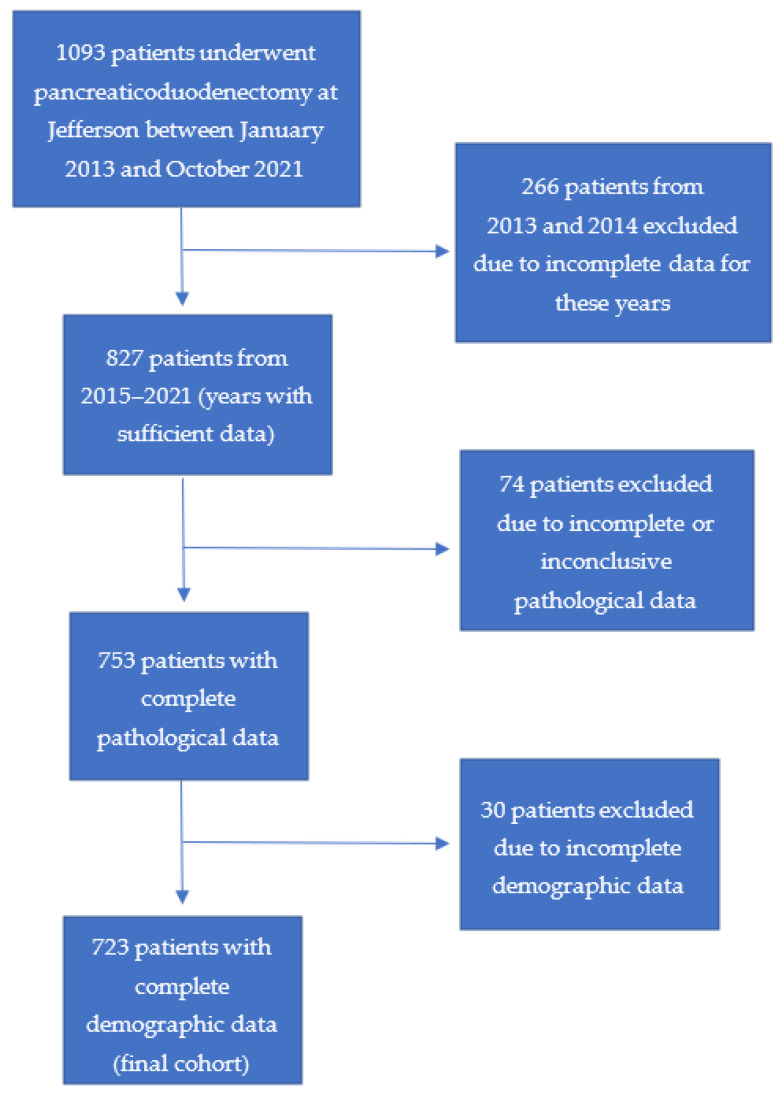
Cohort selection. The initial cohort included 1093 patients from the years 2013–2021, and after exclusion, the final cohort included 723 patients.

**Figure 2 cancers-15-02935-f002:**
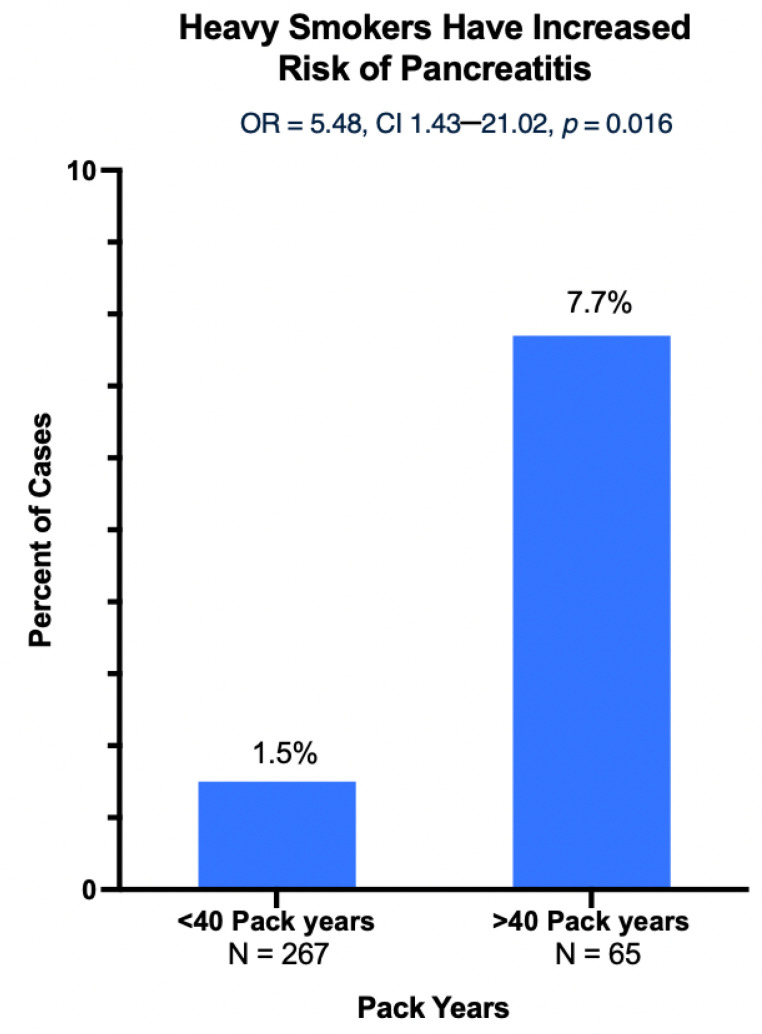
Heavy smokers have increased risk of pancreatitis beyond light/moderate smokers. Note *y*-axis only goes up to 10% for visual aid, but scale is out of 100%.

**Figure 3 cancers-15-02935-f003:**
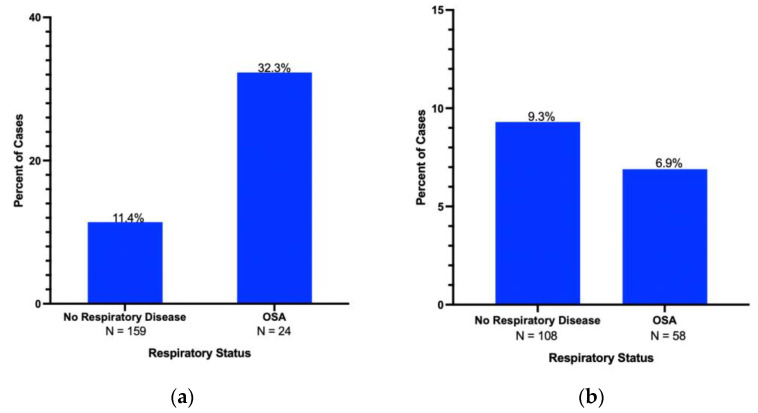
(**a**) IPMN rates in women with OSA (*N* = 24) compared with no respiratory disease controls (*N* = 159) (OR 3.89, 95% CI 1.46–10.37, *p* < 0.01). (**b**) IPMN rates in men with OSA (*N* = 58) compared with no respiratory disease controls (*N* = 108) (OR 0.79, 95% CI 0.25–2.40, P = NS).

**Figure 4 cancers-15-02935-f004:**
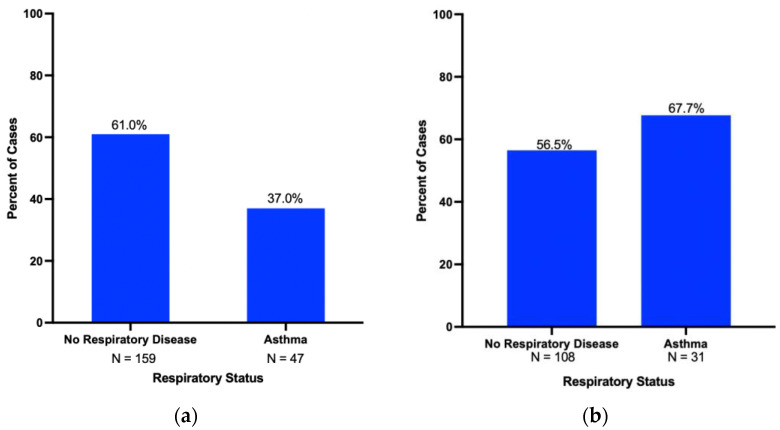
(**a**) Pancreatic and periampullary adenocarcinoma rates in women with asthma (*N* = 47) compared with no respiratory disease controls (*N* = 159) (OR 0.36, 95% CI 0.18–0.71, *p* < 0.01). (**b**) Pancreatic and periampullary adenocarcinoma rates in men with asthma *(N* = 31) compared with no respiratory disease controls (*N* = 108) (OR 1.62, 95% CI 0.96–3.76, P = NS).

**Table 1 cancers-15-02935-t001:** Cohort demographics and respiratory statuses (*N* = 723), 4.7% of patients had more than one single respiratory condition. COPD = Chronic Obstructive Pulmonary Disease, OSA = Obstructive Sleep Apnea, BMI = Body Mass Index.

Total Cases *N* = 723	No Respiratory Disease (NRD) *N* = 267	Smoking History *N* = 382	COPD *N* = 59	OSA *N* = 82	Asthma *N* = 78	Overall N (%)/Mean (SD)
Sex						
Male (*N*=, %)	108 (40.4%)	213 (55.8%)	34 (57.6%)	58 (70.7%)	31 (39.7%)	363 (50.2%)
Female (*N*=, %)	159 (59.6%)	169 (44.2%)	25 (42.4%)	24 (29.3%)	47 (60.3%)	360 (49.8%)
Age (Mean, SD)	66.3 (±12.8)	67.5 (±9.82)	69.9 (±6.80)	66.6 (±8.48)	64.6 (±11.9)	66.8 (±11.25)
BMI (Mean, SD)	26.3 (±5.26)	27.0 (±5.44)	26.2 (±5.03)	29.9 (±5.95)	27.1 (±5.01)	27 (±5.6)
No Respiratory Disease (*N*=, %)	267 (100%)	0 (0%)	0 (0%)	0 (0%)	0 (0%)	267 (36.9%)
Smoking Data						
Positive Smoking History (*N*=, %)	0 (0%)	382 (100%)	51 (86.4%)	48 (58.2%)	37 (47.4%)	382 (52.8%)
Current Smokers (*N*=, %)	0 (0%)	63 (16.5%)	16 (27.1%)	9 (11.0%)	8 (10.3%)	63 (8.7%)
Pack Years (Mean, SD)	0 (±0)	26.1 (±21.0)	33.3 (±24.6)	15.1 (±20.0)	10.4 (±15.4)	23.2 (±20.1)
Years Since Quitting (Mean, SD)	0 (±0)	22.9 (±17.5)	15.92 (±16.6)	27.4 (±14.0)	23.4 (±15.2)	22 (±18.1)
Lung Diseases						
Any Lung Disease (*N*=, %)	0 (0%)	108 (28.3%)	59 (100%)	82 (100%)	78 (100%)	182 (25.2%)
COPD (*N*=, %)	0 (0%)	51 (13.4%)	59 (100%)	16 (19.5%)	13 (16.7%)	59 (8.2%)
Asthma (*N*=, %)	0 (0%)	37 (9.7%)	13 (22%)	15 (18.3%)	78 (100%)	78 (10.8%)
OSA (*N*=, %)	0 (0%)	48 (12.7%)	16 (27.1%)	82 (100%)	15 (19.2%)	82 (11.3%)

**Table 2 cancers-15-02935-t002:** Distribution of pathologies.

Pathology	Number of Cases (*N*=, %)
Pancreatic Ductal Adenocarcinoma (PDAC)	374 (51.73%)
Intraductal Papillary Mucinous Neoplasm (IPMN)	79 (10.93%)
Ampullary Adenocarcinoma	53 (7.33%)
Pancreatic Neuroendocrine Tumor (PNET)	50 (6.92%)
Other Benign	49 (6.78%)
Duodenal Adenocarcinoma	43 (5.95%)
Cholangiocarcinoma	20 (2.77%)
Histologically Proven Chronic Pancreatitis	21 (2.90%)
Adenosquamous Carcinoma	11 (1.52%)
Other Malignant	11 (1.52%)
Solid Pseudopapillary Neoplasm (SPN)	6 (0.83%)
Gastrointestinal Stromal Tumor (GIST)	5 (0.69%)
Sarcomatoid Carcinoma	2 (0.28%)
Acinar Cell Carcinoma	1 (0.14%)

## Data Availability

The data presented in this study are available upon request from the corresponding author.

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
