# Peer review of "Association of Smoking and Respiratory Disease History with Pancreatic Pathologies Requiring Surgical Resection"

_cancers, 2023, doi:10.3390/cancers15112935_

Round 1

Reviewer 1 Report

Thank you for giving me the opportunity to review this retrospective series of patients operated on for pancreatic pathology. 

the authors speak of a control group of 267 patients without respiratory comorbidities. unfortunately, no table is available concerning the characteristics of this subgroup. it would have been interesting to compare the characteristics of patients with or without pulmonary comorbidities. 

one of the objectives of this study was to evaluate the incidence of pancreatic cancer in relation to pulmonary comorbidities. unfortunately, only patients who had undergone surgery were included, which considerably biases the message, since only one out of five patients is eligible for surgical resection. it is necessary to have patients with the same pancreatic pathologies who have not undergone surgery available during the same inclusion period in order to determine the causal relationship. 

Finally, the limitations of the study are not specified. 

Author Response

Please see attached PDF file.

Reviewer 2 Report

Congrats for study. I think the abstract should be better structured. I didn't see anything about alcohol, although you talk a lot about chronic pancreatitis, the cause of which is mainly alcohol consumption. 80-95% of alcohol consumers are smokers, so alcohol also plays a role in the occurrence of pancreatic cancer. I would be interested in how many of the smokers consumed alcohol and the connection with the occurrence of pancreatitis, cancer and IPMN.
 Asthma had a higher risk for developing five types of cancer — lung cancer, blood cancer, melanoma, kidney cancer and ovarian cancer - not a pancreatic cancer. The idea about asthmatics is interesting, but larger studies are needed to draw a conclusion. It is also possible to be a coincidence.

Round 2

Reviewer 1 Report

the authors have answered point by point the questions and comments allowing to significantly improve the manuscript.